# *In silico* discovery and biological validation of ligands of FAD synthase, a promising new antimicrobial target

Isaias Lans[1], Ernesto Anoz-Carbonell[2,3,4], Karen Palacio-Rodríguez[1], José Antonio Aínsa[3,4,5], Milagros Medina[2,3], Pilar Cossio[1,6]*

**1** Biophysics of Tropical Diseases, Max Planck Tandem Group, University of Antioquia UdeA, Medellin, Colombia, **2** Departamento de Bioquímica y Biología Molecular y Celular, Facultad de Ciencias, Universidad de Zaragoza, Spain, **3** Instituto de Biocomputación y Física de Sistemas Complejos (Unidades Asociadas BIFI-IQFR y CBsC-CSIC), Universidad de Zaragoza, Spain, **4** Grupo de Genética de Micobacterias, Departamento de Microbiología, Pediatría, Radiología y Salud Pública. Facultad de Medicina, Universidad de Zaragoza, Zaragoza, Spain, **5** CIBER Enfermedades Respiratorias (CIBERES), Instituto de Salud Carlos III, Spain, **6** Department of Theoretical Biophysics, Max Planck Institute of Biophysics, Frankfurt, Germany

* picossio@biophys.mpg.de

**Data Availability Statement:** All relevant data are within the manuscript and its Supporting Information files.

## Abstract

New treatments for diseases caused by antimicrobial-resistant microorganisms can be developed by identifying unexplored therapeutic targets and by designing efficient drug screening protocols. In this study, we have screened a library of compounds to find ligands for the flavin-adenine dinucleotide synthase (FADS) -a potential target for drug design against tuberculosis and pneumonia- by implementing a new and efficient virtual screening protocol. The protocol has been developed for the *in silico* search of ligands of unexplored therapeutic targets, for which limited information about ligands or ligand-receptor structures is available. It implements an integrative funnel-like strategy with filtering layers that increase in computational accuracy. The protocol starts with a pharmacophore-based virtual screening strategy that uses ligand-free receptor conformations from molecular dynamics (MD) simulations. Then, it performs a molecular docking stage using several docking programs and an exponential consensus ranking strategy. The last filter, samples the conformations of compounds bound to the target using MD simulations. The MD conformations are scored using several traditional scoring functions in combination with a newly-proposed score that takes into account the fluctuations of the molecule with a Morse-based potential. The protocol was optimized and validated using a compound library with known ligands of the *Corynebacterium ammoniagenes* FADS. Then, it was used to find new FADS ligands from a compound library of 14,000 molecules. A small set of 17 *in silico* filtered molecules were tested experimentally. We identified five inhibitors of the activity of the flavin adenylyl transferase module of the FADS, and some of them were able to inhibit growth of three bacterial species: *C. ammoniagenes*, *Mycobacterium tuberculosis*, and *Streptococcus pneumoniae*, where the last two are human pathogens. Overall, the results show that the integrative VS protocol is a cost-effective solution for the discovery of ligands of unexplored therapeutic targets.

**Funding:** This research was founded by Colciencias, University of Antioquia, Ruta N, Colombia, and the Max Planck Society, Germany. Also by the Spanish Ministry of Economy, Industry and Competitiveness (MINECO) [BIO2016-75183-P AEI/FEDER, UE to M.M.], the Spanish Ministry of Science and Innovation (MICINN) (PID2019-103901GB-I00 AEI/FEDER, UE to M.M.] and the Government of Aragón-FEDER [Grupo de Referencia Biología Estructural (E35-20R to M. M.)]. The funders had no role in study design, data collection and analysis, decision to publish, or preparation of the manuscript.

**Competing interests:** The authors have declared that no competing interests exist.

## Author summary

Developing cures for antimicrobial-resistant microorganisms is a pressing necessity. Addressing this problem requires the discovery of novel therapeutic targets -for example, bacterial proteins with no human homologues- and the development of cost-effective drug screening protocols. In this work, we tackled the problem on both sides. We developed an efficient and successful integrative computational protocol for screening inhibitory-molecules for unexplored targets. We used it to discover five novel inhibitors of flavin-adenine dinucleotide synthase (FADS), a promising protein target of pathogens causing tuberculosis and pneumonia.

## Introduction

Given the growing incidence of infections caused by antimicrobial resistant pathogens, international institutions, such as the World Health Organization [1], have informed about the lack of potential therapeutic options for these pathogens, and have named a list of pathogens for which is critical to develop novel antimicrobial agents. The development of such treatments should involve efficient drug design protocols and the discovery of new molecular targets to fight antimicrobial resistance. A straightforward and effective way to increase the chances of success of the drug-screening pipeline is through the implementation of efficient virtual screening (VS) protocols. These methods provide powerful tools to reduce the costs of drug discovery by reducing the number of compounds to be tested in experimental trials [2–5]. Moreover, VS protocols increase the success rate (*i.e.*, active compounds found) and reduce the false negatives in high-throughput compound screening [2, 3, 6–12].

Efficient VS protocols have to be able to screen large compound libraries in short computational times. Therefore, these protocols usually implement a funnel-like strategy, which start from fast but less accurate methods (where a large number of molecules are filtered) and more accurate and time-consuming tools are used in the last steps [13–15]. Usually, pharmacophore-based tools are implemented in the first stages of VS, given their ability to quickly screen large compounds libraries [16–18]. While more sophisticated tools such as docking or molecular dynamics (MD) are implemented in the latter steps to predict ligand affinities [11, 13, 19–26].

Special attention deserve the tools used in the first steps of the VS, because these impact the ability of the protocol to explore large compound libraries and the chemical space of the compounds, such as the pharmacophore-based strategies [16–18, 27–30]. Despite the usefulness of these methods, which accelerate the first steps of the VS, these strategies have some limitations. Several pharmacophore-based methods require knowing ligands or ligand-receptor structures for their training, limiting the chemical space of the filtered molecules to that associated with the training set [16]. Recently, the flexi-pharma method, a VS strategy that uses pharmacophores from MD conformations was developed to overcome these limitations [31]. However, in general, new protein targets, which have limited structural information available, such as the bifunctional enzyme flavin-adenine dinucleotide synthase (FADS), are challenging for the funnel-like VS strategies.

FADS is a potential target for drug design against antimicrobial-resistant organisms, such as the human pathogens *Mycobacterium tuberculosis* and *Streptococcus pneumoniae*. FADS is a bi-functional and bi-modular enzyme that catalyzes the synthesis of two essential co-factors: flavin mononucleotide (FMN) and flavin-adenine dinucleotide (FAD). These are essential for a large number of proteins participating in energy transformation or metabolic processes, in

prokaryotic and eukaryotic organisms [32–34]. For FADS, the synthesis of FMN occurs at the C-terminal module (the RKF module) and that of FAD at the N-terminal module (FMNAT module) [35–38]. The RFK module has a similar structure and sequence to the equivalent enzyme in eukaryotic organisms. However, because the FMNAT module lacks both sequence and structural similarity with the equivalent enzymes in eukaryotic systems, the prokaryotic FADSs have emerged as potential antimicrobial targets [35, 39–41].

The most characterized FADS is the enzyme of *Corynebacterium ammoniagenes* (*Ca*FADS), which is considered a good representative model for the FADS of the human pathogen *M. tuberculosis* (*Mt*FADS) [40, 41]. However, limited structural information about this enzyme is available. Moreover, no experimental structures of the FADS-FMNAT module in complex with substrates are reported. The only ligand-*Ca*FADS FMNAT module structures correspond to theoretical or computational models [35, 42].

The objective of this work is to discover molecules able to inhibit the FMNAT activity of FADS using a novel integrative VS protocol. This promising protocol addresses some of the limitations found with traditional VS, for example, it does not require knowledge of ligands or ligand-receptor structures, and its attributes enable a better exploration of the chemical space of large compound libraries.

This manuscript is organized as follows. First, we describe the integrative computational protocol, which includes several filtering layers: *i*) flexi-pharma screening [31], *ii*) consensus docking screening [43], *iii*) MD sampling and scoring, and *iv*) compound activities measured by experimental assays. The protocol is tested and optimized using a library of 1993 compounds from which 39 compounds are true ligands of the *Ca*FADS [41]. Subsequently, the optimized protocol is implemented over a library of 14000 compounds. A final list of 17 filtered compounds is tested experimentally. We discover that six molecules are able to inhibit the FMNAT-FADS activity, five bind to the FMNAT-FADS and five present growth inhibitory activity against *C. amoniagenes*, *M. tuberculosis* or *S. pneumoniae*. We conclude that the VS protocol and the new inhibitory compounds can contribute to further development of novel therapeutic strategies against antimicrobial-resistant pathogens such as *M. tuberculosis* and *S. pneumoniae*.

## Materials and methods

### Computational methods

**FAD structure.**   The Cartesian coordinates of the FMNAT module (M1-H186) of *Ca*FADS were taken from the crystal structure with PDB code 2X0K [35]. These were used for the MD simulations in the flexi-pharma or molecular docking stage.

**Flexi-pharma method.**   The flexi-pharma protocol [31] has three substages: run an MD simulation of the receptor target, generate a set of pharmacophores from each ligand-free receptor MD conformation, and assign a vote to each molecule every time it matches at least one pharmacophore from each MD conformation.

First, an MD simulation of receptor was performed. For *Ca*FADS, we used the results from a simulation of this system performed in a previous work [42]. Specifically, we used 600 equidistant ligand-free *Ca*FADS conformations from 60 ns of MD at 300.15K (for details about the MD parameters see ref. [42]). To generate the pharmacophore set from each ligand-free MD conformation, we used Autogrid4.2 [44] to calculate the affinity maps of several atom-types: hydrogen-bond donor, hydrogen-bond acceptor, hydrophobic, aromatic and charged atoms. Some atom-type affinity grids were first discarded if they show a flat distribution of the affinity values of the map (for this work, the affinity maps that have a histogram with kurtosis larger than 3). Then, we defined a grid-percentage threshold to determine the hotspots (clusters of

grid-cells) for each atom type. The threshold is a percentage of the total number of cells in the grid with negative affinity energy. We clustered the selected cells to generate a pharmacophoric feature (given by a center, a radius of gyration, an atom type and in some cases a direction). A pharmacophore was built by combining three features. The pharamcophore set consists of all possible combinations of triplets of features from different active spaces (*i.e.*, centers of the affinity grids) together with a volume exclusion term. The Pharmer [45] program was used to screen the compound library with the created pharmacophore set. An example of the pharma-cophore mapping is shown in S1 Fig.

Flexi-pharma gives a score to each compound by means of a voting strategy. If a molecule, from the compounds library, matches any pharmacophore obtained from a specific MD frame, then the molecule obtains a vote. The total number of votes is used as a score of the molecule. For more details about the flexi-pharma method see ref. [31].

**Molecular docking.** The molecular docking was carried out using the programs: Auto-dock4.2 [44, 46, 47], Vina [48], and Smina [49]. All programs used the same molecule input format that was AutoDock pdbqt. The protonation state for each molecule was determined from the compound libraries. Also for these programs, the sampling space was defined using a grid box of $15 \times 15 \times 15$ Å centered at the oxygen of the amide group of the catalytic residue ASN125 [50]. The number of requested poses was 50.

Autodock4.2 [44, 46, 47] was used with a grid spacing of 0.25 Å. The search was performed using the Lamarckian genetic algorithm implemented in Autodock, with a starting population of 50 individuals, using 25000000 energy evaluations and 27000 generations. The resultant poses were clustered using the RMSD of the atomic positions, with a tolerance of 2.0 Å, using the default clustering method. In addition, to the sampling space and the number of poses, Vina [48] and Smina [49] were used with the default parameters. For Smina, the Vinardo [51] scoring function was used.

**Molecular dynamics simulations.** The best pose for each compound from the docking stage, obtained with the Autodock4.2 program, was used as the initial conformation for the MD simulation. Since, the output poses from Autodock4.2 do not contain aliphatic protons, we use Open Babel [52] to protonate those atoms as in the original database (Prestwick or Maybridge). The PROPKA [53] module from the PDB2PQR software package [54, 55] was used to determine the protonation state of all ionizable groups at pH 7.0. The final models were solvated with a dodecahedral water box, centered at the geometric center of the complex. To neutralize the systems, $Na^+$ ions were added when necessary. The AMBER99SB-ILDN [56] force field was used to model the protein with the TIP3P water model [57]. The GAFF force field [58] parameters were obtained for the compounds using Antechamber [58, 59]. ACPYPE [60] was used to change the topology files from amber to GROMACS [61, 62], which was used for all the MD simulations. The systems were minimized until the maximum force was $\leq 1000$ kJ/mol·nm with the steepest descent algorithm. MD simulations were carried out with periodic boundary conditions. A spherical cutoff of 1.2 nm for the non-bonded interactions was applied together with a switch function acting between 1.0 and 1.2 nm. The non-bonded pair list was updated every 20 steps. The particle mesh Ewald method was used to compute long-range elec-trostatic force terms, and the leapfrog algorithm to propagate the equations of motion. All bond lengths and angles involving hydrogen atoms were constrained using the LINCS algo-rithm [63]. Equilibration consisted of 100 ps of NVT followed by 100 ps of NPT simulation at 310 K, with a time step of 2 fs. During equilibration the coordinates of protein and of ligand heavy atoms were restrained using a constant force of 100 kJ/mol·nm. Finally, MD simulations between 5—15 ns were carried out using the GROMACS 5.1.3 program [61, 62] with a time step of 2 fs, without restraints, in an isothermal-isobaric (NPT) ensemble at 310.15 K and 1 atm.

**Exponential consensus ranking.** A consensus methodology was used to combine the results from different scoring functions both for the docking and MD VS stages. We used an exponential consensus ranking (ECR) methodology [43]. This method assigns a score $p(r_i^j)$ to each molecule $i$ for each scoring function $j$ using an exponential function $p(r_i^j) = \exp\left(\frac{-r_i^j}{\alpha}\right)/\alpha$, which depends on the rank of the molecule $(r_i^j)$ given by each individual docking program. $\alpha$ is the expected value of the exponential distribution, which we have set to 50% of the total molecules at each stage. The final score $P(i)$ is defined as the sum of the exponential functions for all of the programs, $P(i) = \sum_j p(r_i^j) = \frac{1}{\alpha}\sum_j \exp\left(\frac{-r_i^j}{\alpha}\right)$.

**Validation metrics.** The enrichment factor (EFx%) is a measure of the change on the ligand/decoys proportion in a molecular dataset, after filtering it to the x%. EFx% is defined as the ratio between ligands (Hits) found at a certain threshold (x%) of the best ranked compounds and the number of compounds at that threshold (Nx%) normalized by the ratio between the hits contained in the entire dataset (Hits100%) and the total number of compounds N100%:

$$EF_x = \frac{Hits^{x\%}}{N^{x\%}} \times \frac{N^{100\%}}{Hits^{100\%}}. \tag{1}$$

Values of EFx% higher than 1 indicate an enrichment of the compound library.

The enrichment plot (EP) measures the performance of a filtering method at different levels of a compound library reduction. In an EP, the percentage of ligands found in the top x% of ranked compounds vs the top x% of filtered compounds is plotted [64].

To asses to the error of the flexi-pharma EPs, a bootstrapping analysis with replacement was used. The selected MD frames were iteratively re-sampled with replacement 100 times. Thus, 100 EPs were obtained for each trajectory. From these the average and the error of the EPs were calculated (similarly as in ref. [31]).

## Experimental methods

**Chemicals.** The selected compounds were acquired from Molport and dissolved in 100% DMSO to prepare stock solutions at 50 mM and 10 mM. According to the manufacter indications, the purity of the compounds was >95%, and had been determined by high performance liquid chromatography (HPLC), thin layer chromatography (TLC), NMR, IR or basic titration.

**Protein purification and quantification.** *Ca*FADS was produced as a recombinant protein in *Escherichia coli* BL21(DE3) and purified as previously described in ref. [40]. Protein purity was tested by 15% SDS-PAGE. Protein content in pure samples (in 20 mM PIPES, pH 7.0) was quantified using the theoretical extinction coefficient ($\epsilon$) 279 nm = 27.8 mM$^{-1}$·cm$^{-1}$.

**Differential scanning fluorescence.** Interaction of compounds with *Ca*FADS was evaluated using fluorescence thermal denaturation, on the bases of the shifts in denaturation midpoints of thermal curves of the protein [65]. Denaturations were performed in a Stratagene Agilent Mx3005p qPCR instrument (Santa Clara, US) following SYPR Orange (ThermoFisher Scientific) emission fluorescence (excitation at 492 nm and emission at 610 nm), which greatly increases when this probe binds to protein hydrophobic regions becoming solvent exposed upon thermal unfolding. Solutions containing 2 $\mu$M *Ca*FADS with the studied compound in an increasing 5-250 $\mu$M concentration range (2% residual final concentration of DMSO) and 5xSYPR Orange in 20 mM PIPES pH 7.0, 10 mM MgCl$_2$, with a 100 $\mu$L total volume were dispensed into 96-well microplates (BRAND 96-well plates pure grade™). After an initial 1 min incubation at 25 ˚C within the equipment, unfolding curves were registered from 25 to 100 ˚C

at 1 °C·min$^{-1}$. Control experiments with *Ca*FADS samples with/without DMSO were routinely performed in each microplate. For those compounds shifting midpoint denaturation temperature, $T_m$, the dissociation constant, $K_d$, was predicted by fitting the data to the equation [66]

$$\frac{\Delta T_m}{T_m} = \frac{NRT_m^0}{\Delta H_0} \ln\left(1 + \frac{[L]}{K_d}\right), \tag{2}$$

which estimates the extent of the ligand-induced protein stabilization/destabilization. $\Delta T_m = |T_m - T_m^0|$ with $T_m^0$ and $T_m$ being the midpoint denaturation temperatures in the absence and the presence of ligand, respectively, and $\Delta H_0$ the unfolding enthalpy of the protein in the absence of ligand.

**Evaluation of the compound's ability to inhibit the *Ca*FADS enzymatic activity.** To determine the compound's ability to inhibit the RFK and/or the FMNAT activities of *Ca*FADS, both enzymatic activities were quantitatively measured in the absence and presence of the compounds following previously described protocols [41]. Reaction mixtures contained 50 $\mu$M ATP, 5 $\mu$M RF in 20 mM PIPES, pH 7.0, 0.8 mM MgCl$_2$, when assaying the RFK activity, and 50 $\mu$M ATP, 10 $\mu$M FMN in 20 mM PIPES, pH 7.0, 10 mM MgCl$_2$ when measuring the FMNAT reaction. Each compound was tested at 250 $\mu$M (0.5% residual final concentration of DMSO) for each of the two enzymatic reactions. The samples were pre-incubated at 25 °C, the reaction was then initiated by the addition of $\sim$ 40 nM *Ca*FADS (final concentration) and allowed for 1 min. Finally, the reaction was stopped by boiling the samples for 5 min and the denatured protein was eliminated through centrifugation. The transformation of RF into FMN and FAD (RFK activity) and of FMN into FAD (FMNAT activity) was evaluated through flavins separation by HPLC (Waters), as previously described [37]. All the experiments were performed in triplicate. To evaluate the potency of compounds as inhibitors, we took advantage of the decrease in quantum yield of fluorescence when FMN is transformed into FAD, which allows to follow such transformation in a continuous system. Measurements were carried out using a microplate reader Synergy HT multimode plate reader (Biotek) with BRAND 96-well plates pure Grade. Reaction mixtures contained 5 $\mu$M RF or FMN, and 50 $\mu$M ATP in 20 mM PIPES, pH 7.0, 10 mM MgCl$_2$, and the inhibitor compound in a 5-250 $\mu$M concentration range (2% residual final concentration of DMSO). Reactions were initiated through addition of 0.4 $\mu$M CaFADS, being the final reaction volume 100 $\mu$L. Flavin fluorescence (excitation at 440 nm and emission at 530 nm) was registered at 25 °C, every 50 s during 15 min. The fluorescence change per time unit ($\Delta$F/$\Delta$t) was calculated as the slope of the resulting fluorescence decays recorded between 0 and 6 min (linear decay of the fluorescence). Controls which contained the reaction mixture without the enzyme and without any potential *Ca*FADS inhibitory compound were included in the assay and referred as the 0% and 100% of enzymatic activity, respectively. IC$_{50}$ was calculated as the concentration of compound required for a 50% inhibition of the enzymatic activity.

**Determination of the antibacterial activity of the compounds.** The minimum inhibitory concentration (MIC) of the inhibitors was determined by the resazurin serial broth microdilution method [67] according to the Clinical and Laboratory Standards Institute guidelines. Compounds were tested against a panel of bacterial strains including Gram positives, Gram negatives and acid fast bacteria (see S3 Table). Serial 2-fold dilutions of the inhibitors were performed in cation-adjusted Mueller-Hinton broth (Difco) in 96-well polypropylene flat-bottom plates, with a final volume of 100 $\mu$L per well. Subsequently, liquid cultures of the bacterial strains in logarithmic phase were adjusted to 10$^6$ CFU/ml in Mueller-Hinton broth, and 100 $\mu$L of this suspension were added to each well, resulting in a final inoculum of 5·10$^5$ CFU/ml.

Plates were incubated for 18 hours at 37 ˚C. Then, 30 $\mu$L of 0.4 mM filter-sterilized resazurin (Sigma-Aldrich) was added to each well, and results were revealed after 4 h of further incubation at 37 ˚C. When testing the compounds against mycobacteria, Middlebrook 7H9 (Difco) supplemented with 10% ADC (0.2% dextrose, 0.5% V fraction BSA and 0.0003% bovine catalase) (BD Difco) and with 0.5% glycerol (Scharlau) was used as culture media, and plates were incubated 4 days for *Mycobacterium smegmatis* and 7 days for *M. tuberculosis*. Resazurin (blue) is an indicator of bacterial growth, since metabolic activity of bacteria reduces it to resorufin (pink). The minimum inhibitory concentration (MIC) is the lowest concentration of compound that does not change the resazurin colour from blue to pink.

**Evaluation of the cytotoxicity of the compounds in eukaryotic cell lines.** The (4,5-dimethylthiazol-2-yl)-2,5-diphenyl tetrazolium bromide (MTT) assay was used to determine the effect of the compounds in cell growth and viability of HeLa (ATCC CCL-2) and A549 (ATCC CCL-185) eukaryotic cell lines. Both cell lines were routinely cultured in high-glucose DMEM (Lonza) supplemented with 10% fetal bovine serum, 4 mM glutamine GlutaMAX™ (Gibco) and 1x non-essential amino acids (Gibco), under 5% $CO_2$ at 37 ˚C in a humidified atmosphere. For the MTT assay, cells were initially seeded in 96-well flat-bottom plates at a density of $10^4$ cells per well and cultured for 24 h. Cultures were routinely tested for mycoplasma presence. The compounds were dissolved in fresh culture medium, added in a 4-512 $\mu$M concentration range (1% DMSO final concentration), and incubated with the cells for 24 h. Finally, formazan crystals were dissolved with pure DMSO and MTT absorbance was measured at 570 and 650 nm. Untreated cells were included as control of 100% viability. Assays were done in quadruplicate.

# Results and discussion

## Virtual screening protocol

The VS protocol aims to find active compounds, from large compound libraries, towards receptors for which little or no information about ligands or ligand-receptor structures is available. This is the case of the *Ca*FADS. To achieve this goal, we implemented a funnel-like protocol with four filtering stages (Fig 1). It includes three main VS stages plus an experimental stage. In the following, we present the principal ideas for the integrative VS protocol.

**Flexi-Pharma: Pharmacophore filtering from ligand-free receptor conformations.** Pharmacophore-based VS strategies are computationally efficient. These strategies are able to explore large compound libraries using pharmacophores: an ensemble of physico-chemical features that ensure the optimal interactions within the active site of a specific biological target [16]. Therefore, the first stage of the protocol implements a phamacophore-based VS strategy [31]. The method flexi-pharma defines pharmacophores from ligand-free receptor conformations from MD simulations. It implements a rank-by-vote strategy, assigning a vote to each compound that matches an MD conformation. The use of multiple conformations allows for a better exploration of the pharmacophoric space. The voting strategy enables the filtering of the molecules at any percentage of the dataset. Details for the flexi-pharma strategy are presented in the Methods and in ref. [31].

**ECR-docking: Exponential consensus ranking of docking VS.** The second stage of the protocol consists of a docking-based VS. Molecular docking aims to find the most favorable binding conformation of a molecule (*i.e.*, pose) upon binding to a pocket of a protein target [68, 69], and assigns a docking score to each molecule. The docking score is an empirical or physics-based estimation of the affinity of the molecule towards the biological target. Therefore, with molecular docking, it is possible to screen and rank molecules from compound libraries. However, it has been shown that the docking results might be system or structure

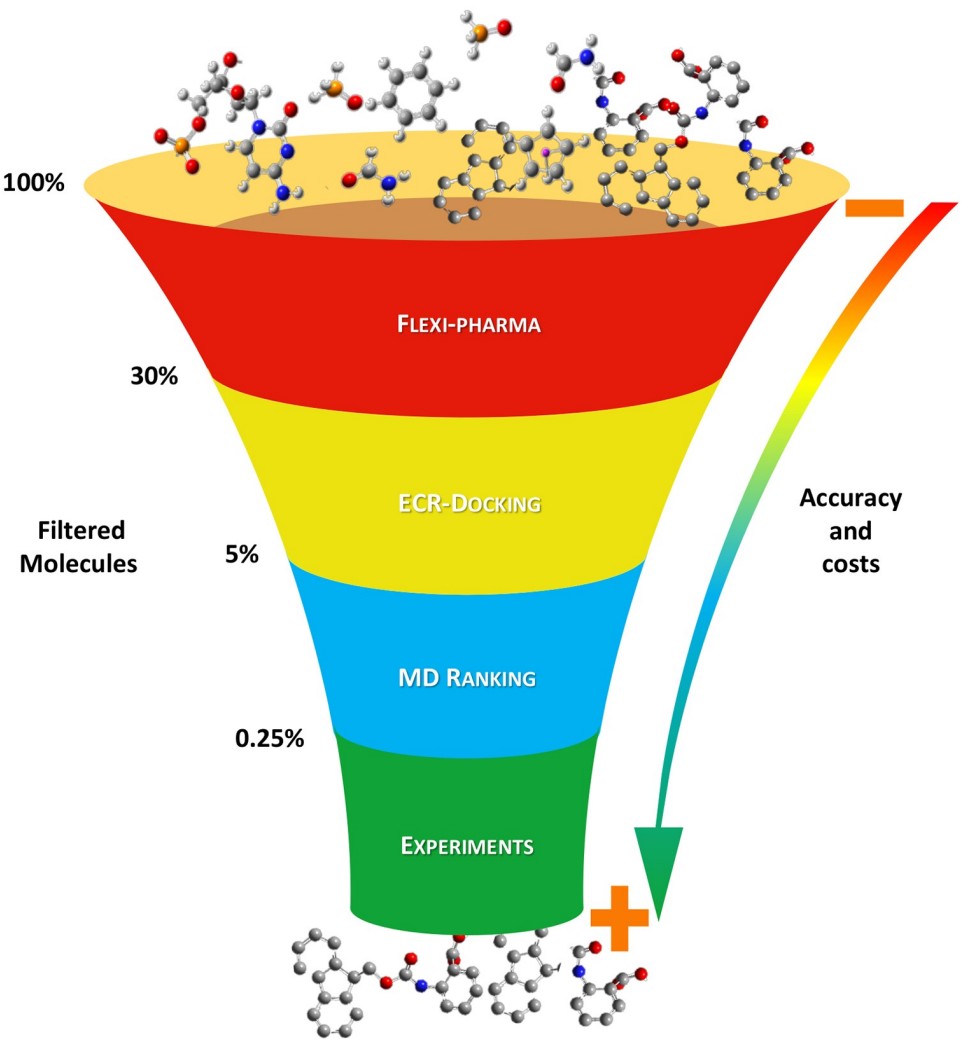

**Fig 1. Main stages of the funnel-like VS protocol.** The protocol consists of four stages: *i*) flexible pharmacophore-based VS (flexi-pharma) [31], *ii*) docking and exponential consensus ranking (ECR-docking) [43], *iii*) MD simulations with consensus ranking -that includes a new Morse-based ligand-flexibility score-, and *iv*) biological experimental binding and activity assays. At each stage, the compound library was filtered. The protocol was optimized and validated over a library of 1993 compounds, which was previously tested over *Ca*FADS [41]. On the left, we show an example of the reduction of this library going from 100% to 0.25% through the successive steps.

dependent [43, 70], possibly due to algorithm-parameterization biases, which are trained over particular benchmark systems. To overcome this limitation, we use a consensus strategy that combines the results from different docking programs to obtain a consensus rank using a sum of exponential functions (ECR method) [43]. Below, in the S1 Text and S2 Fig, we describe the different docking-scoring alternatives that we used to find the optimal enrichment for the FMNAT-FADS ligand screening.

**MD-ranking VS.** MD simulations were used to estimate the compound affinities and the stability of the predicted complexes filtered from the docking stage. The MD starting configuration was selected from the best pose obtained with Autodock4.2 [44, 46, 47] in the previous stage. Inspired by conformational-prediction tools that take into account flexibility [71, 72], we used two measures for the stability and affinity of the ligand bound to the receptor in the MD ensemble. The first measure generates a consensus rank using multiple scoring functions

over the MD conformations. We called this scoring function-based rank (see below for details). The second measure is based on the root mean square deviation (RMSD) of the ligand's atomic positions along the MD trajectory. This is used with a Morse potential to define a score that measures the ligand's flexibility (see below for details). Finally, the scoring function-based rank and the Morse-based rank are combined using the ECR method. We use this analysis to select the percentage of best-ranked molecules for the activities assays. In the following, we describe the scoring function-based rank and the Morse-based rank.

**Scoring function-based rank.**   We used four scoring functions: Autodock4.2 [44, 46, 47], Vina [48], Vinardo [51] and CY score [73], which were calculated over each MD conformation. For each scoring function, an average score over all the conformations is calculated for each molecule. This can then be used to rank the molecules. The ranks from the four scoring functions are used with the ECR method [43] to obtain a consensus rank by combining their individual ranks.

**Morse-based rank.**   We used the standard deviation of the RMSD of the ligand's atomic positions around the binding site as an indicator of the ligand's flexibility. Those ligands showing a small standard deviation of the RMSD at the binding site indicate a very rigid complex, which leads to a conformational penalization. On the other hand, molecules with high RMSD standard deviation, indicate a dissociation tendency, which leads to an affinity penalization. These behaviors can be characterized using a Morse potential (Eq 3 and Fig 2)

$$V_M(r) = \omega(1 - \exp^{-a(r-r_0)})^2, \tag{3}$$

where $\omega$ is the depth of the well, $r_0$ is the position of the minimum, $a = \sqrt{\frac{k}{2\omega}}$ and $k$ is a constant that defines the width of the well. For the Morse-base score, we used $V_M(r)$ from Eq 3, where the dependent variable $r$ is the standard deviation of the RMSD along the MD trajectory, and $r_0$ is the standard deviation corresponding to a normal distribution with null entropy (*i.e.*, $r_0 = 0.242$ Å). The Morse potential was implemented with a force constant $k = 1$ kcal/mol.nm$^2$ and a depth of the well $\omega = 1$ kcal/mol. Thus, RMSD values lower or higher than 0.242 Å are penalized with Morse-based score. We used this to rank the molecules according to the $V_M(r)$ score. We note that most molecules have a RMSD standard deviation greater than 0.242 Å, therefore, the parameters $k$ and $\omega$ used in the score do not have a great impact in the final Morse-based rank.

**VS parameter dependence.**   Although the presented VS protocol is sufficiently general to be applied over any receptor target, there are several parameters and setups that can be optimized. Moreover, because -in its complete form- it has not been tested, we considered it necessary to first validate and optimize the VS protocol over a benchmark library with known inhibitors of the *Ca*FADS—FMNAT activity [41]. The results are presented in the following section.

## VS protocol FADS validation: Prestwick Chemical Library

To validate and optimize the VS protocol for screening potential ligands of *Ca*FADS, a molecular library (Prestwick Chemical Library) was used. A previous study showed that 39 of its 1993 compounds are able to bind to the *Ca*FADS with FMNAT inhibitory activity [41]. To study the performance, we measure the enrichment factor (EF) and the enrichment plots (EP) [64] (see the Methods). In the following, we present the results for each stage of the VS protocol applied over the Prestwick Chemical Library.

**Flexi-pharma FADS optimization and validation.**   In Fig 3, we present the EP obtained after the application of the flexi-pharma stage over the Prestwick compound library. Since the

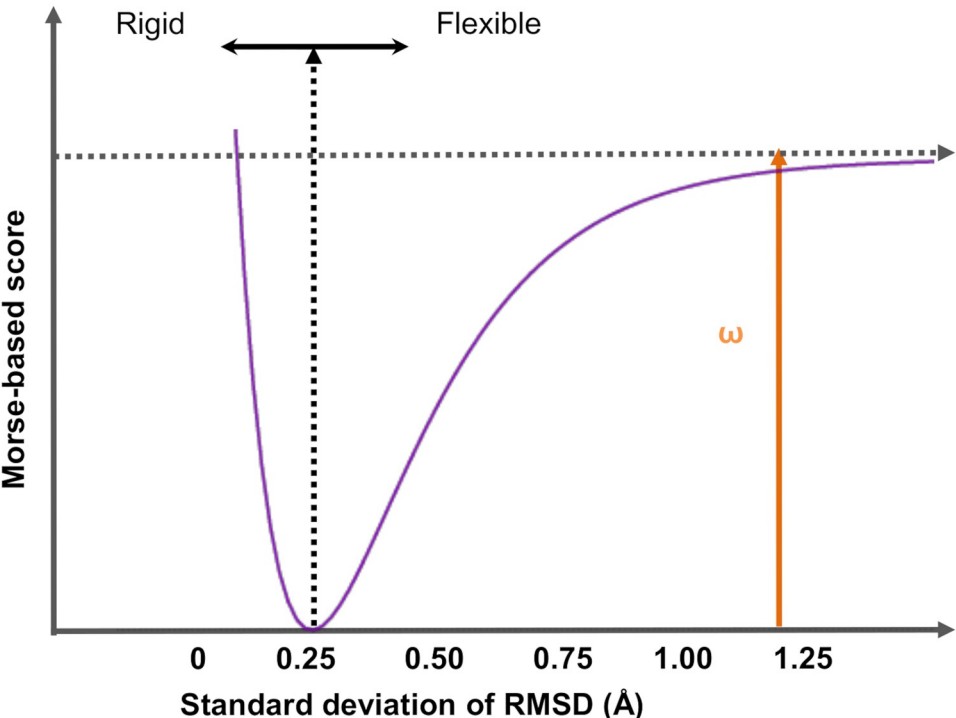

**Fig 2. Morse-based score.** A score that uses a Morse potential (Eq 3) was implemented for scoring the flexibility of the ligand inside the pocket using MD simulations. The input variable is the standard deviation of the RMSD of the ligand's atomic positions around the binding site. Ligands that show large RMSD variations are considered very flexible -with dissociation tendencies (*i.e.*, unstable)- and their behavior is penalized (right of vertical black dashed arrow). Ligands with small RMSD fluctuations are considered rigid leading to a conformational penalization (left of black vertical dashed arrow). The Morse potential was implemented with a force constant $k = 1$ kcal/mol.nm$^2$, a depth of the well of $\omega = 1$ kcal/mol, and the minimum is localized at $r_0 = 0.242$ Å.

flexi-pharma method uses MD conformations of the ligand-free receptor, we used 600 equidistant frames from 60 ns of MD of the ligand-free *Ca*FADS, which was carried out in a previous study [42]. We applied the flexi-pharma VS as described in the Methods. We find an enrichment of the compound library, showing that the data (black line) are better than a random EP (red line). The vertical violet line shows the percentage of molecules selected to pass to the next stage. The selected list of compounds consists of 600 potential ligands ($\approx$30% of the initial compound library) with 24 actual ligands, resulting in a EF of 2.0 for this stage. These results support the usefulness of flexi-pharma to enrich the Prestwick compound library.

The EPs showed in the Fig 3 involved several parameters, such as the affinity grid threshold value (*i.e.*, percentage of grid points with lowest grid energies), the active spaces and the number of features used to define the pharmacophores (see ref. [31]). In that work, it was shown that the results are almost independent of the affinity-grid threshold. However, a large threshold implies a large number of features, which increases the pharmacophore set and the computational time to carry out the VS. Therefore, a good computational efficiency is obtained with small threshold values, while maintaining the performance. For this study, the threshold value of 0.1% is used. Because of the large size of the FMNAT active site, the pharmacophores were obtained from 7 active spaces (centered at NE2-H31, NE2-H57, CA-E108, CG-L110, ND2-N125, OG-S164 and CZ-R168 [50]).

**ECR-docking FADS optimization and validation.** The second stage of the VS protocol uses a docking-based strategy. Docking generates an optimal molecule-bound conformation

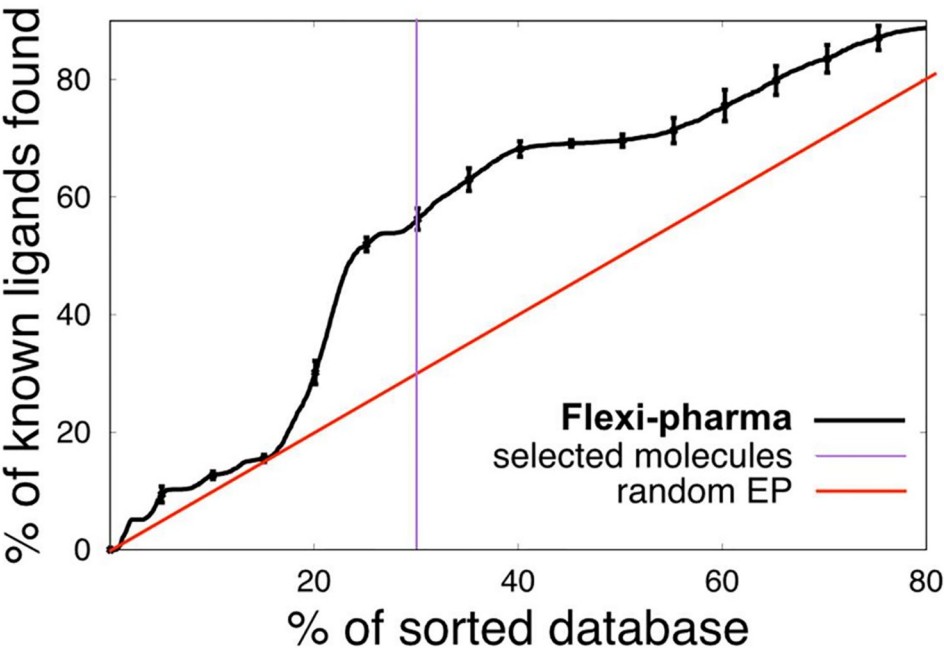

**Fig 3. Flexi-pharma VS stage over the Pretswick library.** Average enrichment plot of the Pretswick library using the flexi-pharma stage over MD conformations of ligand-free *Ca*FADS. The affinity-grid threshold value is 0.1% and 600 equidistant frames obtained from an MD of 60 ns were used [42]. The flexi-pharma number of votes for each molecule was used as a score to calculate the EPs. Bootstrapping analysis was performed by sampling with replacement 100 times to obtain the average EP and its standard deviation. The violet line shows the screening threshold (≈30%) for the selection of molecules to be filtered and passed onto the second stage.

with a corresponding score. However, some docking-program outcomes depend on the system of study [43, 70]. Thus, a particular docking software can show good results for a receptor, however, it can show bad results for other receptors. For an untested receptor it is impossible to know, in advance, which docking software generates the best outcome. To overcome this limitation, we implement a modified version of the exponential consensus rank (ECR) strategy [43] using several docking programs.

The top 600 molecules of the Pretswick library filtered from the flexi-pharma stage were docked, using several programs, to the FMNAT module using the crystallographic structure (PDB 2X0K) of *Ca*FADS. After several attempts (see the S1 Text), we found that the best EP was obtained by using the best pose from Autodock4.2 and re-scoring it with Autodock4.2 [44, 46, 47], Vina [48], Vinardo [51] (a function scoring implemented in Smina [49])) and CYscore [73] scoring functions. The molecules that did not have the best Autodock4.2 pose within the FMNAT-FADS active site (less than 5 Å of H31 and N125) [50] were discarded, reducing the list to 467 molecules (including 23 confirmed inhibitors). The ranks from each scoring function were combined in using an ECR methodology to obtain a consensus rank. The enrichment plot after this analysis is shown in the Fig 4 and S2 Fig. The top 100 molecules from this analysis contained 9 confirmed ligands, which represents a global EF% (*i.e.*, the EF% normalized to the initial compound library) of 4.6, showing a clear enrichment.

**MD-ranking FADS optimization and validation.** MD simulations of potential ligand-receptor complexes allows for a more accurate sampling of their conformational space. The

## ECR-docking VS stage

**Fig 4. ECR-docking VS stage over the Pretswick library.** Enrichment plot using the ECR (black line) from the best Autodock4.2 pose that is re-scored with four scoring functions (Autodock4.2, Vina, Vinardo and CYscore). The shaded area encloses the best and worst behaviors for the individual scoring functions. The enrichment plot is normalized by the initial database values (39-ligands and 1993 compounds). The violet line shows the threshold for the selection of molecules for the third VS stage.

stability of these conformations should give an estimate of the affinity of the potential ligand towards the receptor. As mentioned previously, the starting conformation was chosen from the best docking pose from Autodock4.2 obtained from the previous stage. For the Pretswick library, we found that by dividing the MD stage into two substages according to the simulation time (MD-ranking 1 and MD-ranking 2), there is a good trade-off between computational costs and performance. In the MD-ranking 1, we ran 5 ns for all the compounds filtered from the docking stage. We used the two measures over the MD conformations, scoring function-based and Morse-based, to rank of the potential ligands to the *Ca*FADS. The two results were combined using an ECR to obtain the EPs shown in Fig 5 (black line). Our results indicate that it is possible to obtain an enrichment of the library compound using MD and combining the two measures using an ECR methodology. We note that, although the EP from the Morse-base rank shows better outcome than the EP from the ECR scoring functions-based (ECR-SF) (blue and green lines, respectively, in Fig 5), the use of both measures is relevant. The logic behind combining the two stability measures, lies in that the Morse-based rank gives only information about the conformational stability of the ligand but it does not contain direct information about any physico-chemical interactions. Whereas the scoring functions, *e.g.*, Autodock4.2 or Vina, include physics-based interactions which are relevant. Therefore, the scoring functions-based rank supplies an empirical contribution to the enthalpy and global entropy in the binding process. Thus, the two strategies should complement each other.

We selected the top 50 molecules to be screened in the MD-ranking 2 stage. For this filtered set, we extended the MD time to 15 ns for each complex. These new conformations are scored

**Fig 5. MD-ranking 1 VS stage over the Pretswick library.** Enrichment plot obtained using an ECR methodology (black) from the combination of the ECR scoring function-based rank (ECR-SF) (green) and Morse-based rank (blue). 5 ns of MD for 100 complexes were carried out. We used 200 equidistant frames from the last 2 ns of MD simulation. The enrichment plot is normalized to the initial database (39-ligands and 1993 compounds). The violet line shows the threshold for the molecule selection for the MD-ranking 2 stage.

similarly as before (*i.e.*, ECR combined scoring function-based and Morse-based ranking) and we select the top 5 molecules. The EFs for all the stages of the protocol are shown in Table 1. These results confirm that all stages of the VS protocol increase the enrichment while saving computational resources.

## VS protocol FADS application: Maybridge database

Once the efficiency of the VS protocol was optimized with the Prestwick library, the protocol was implemented over the Maybridge compound library that contains 14000 molecules. A description of the VS protocol for this library is shown in S3 Fig. For the flexi-pharma filtering we selected 3000 molecules. Then, the best 600 were selected using the ECR-docking stage. For the second MD-ranking VS stage, we tested the best 300 molecules. In S4 Fig, we present the

**Table 1. EFs obtained for each VS stages using the Prestwick library.**

| Stage | Stage EF | Global EF | Ligands | Molecules filtered |
|---|---|---|---|---|
| Flexi-pharma | 2.0 | 2.0 | 24 | 600 |
| ECR-Docking | 2.3 | 4.6 | 9 | 100 |
| MD-ranking 1 | 1.3 | 6.1 | 6 | 50 |
| MD-ranking 2 | 1.7 | 10.2 | 1 | 5 |

Stage EF is the enrichment factor relative to the previous step, while the Global EF is the enrichment factor normalized to the initial database (39-ligands and 1993 compounds).

rank obtained from each individual scoring function in comparison to the ECR, which was used to select the top 30 molecules. We note that the difference in the screening between the Maybridge library with respect to the Prestwick library is the percentage of molecules selected at each step. These numbers were changed to optimize the computational efficiency, since for just the MD-ranking stages 6 $\mu$s of MD simulation time were used.

## Experimental evaluation of the affinity and the inhibitory activity of selected molecules

Considering a range of properties for the 30 best VS-ranked compounds that relate to their potential drug-likeness, shown in S1 Table [74, 75], as well as their commercial availability, 17 compounds were chosen as virtual screening hits (VSH) to experimentally evaluate their performance. From the experimental assays 5 compounds were found as true ligands of *Ca*FADS. Table 2 and Fig 6 show the dissociation constant ($K_d$) values in the range of 1.7—41 $\mu$M.

Since binding of small molecules to a protein usually alters its thermal conformational stability, shifting the midpoint temperatures ($T_m$) of thermal denaturation curves [65], displacements in $T_m$ induced by the different VSH appeared as a feasible approach to experimentally identify those binding *Ca*FADS [76]. 5 compounds, out of the 17 selected, produced a dose-response $T_m$ shift, $\Delta T_m$, indicative of interaction with *Ca*FADS (Fig 6A). Compounds C6 and C9 increased $T_m$ by more than 3 degrees, indicating binding to the protein. In addition, C3, C5 and C18 shifted it to lower values (up to 2 and 6 degrees, respectively), suggesting that they produced a ligand-induced perturbation consistent with binding and destabilization of *Ca*FADS. Fitting of the corresponding dose-response data to Eq 2 that relates them to the binding affinity, allowed to estimate the corresponding $K_d$ values (Fig 6B, second column in Table 2). The data pointed to C5, C9 and C18 as the stronger binders. In addition, postulated the C5 >C18 >C9 >C3 >C6 affinity ranking with $K_d$ values in the 1.7-41 $\mu$M range. The previous results support that compounds C5, C6, C9, C18 and C3 are actual ligands of *Ca*FADS, highlighting the capacity of the VS protocol to find protein ligands for receptor targets.

As our VS was directed towards the FMNAT active site of *Ca*FADS, we then rated the power of the 17 VSHs as inhibitors of *Ca*FADS ability to transform FMN into FAD. Hits were evaluated in terms of concentration of compounds causing 50% enzyme inhibition ($IC_{50}$), as

**Table 2. *In vitro* performance of VS hits over the FMNAT and RFK+FMNAT *Ca*FADS activities.**

| Compound | $K_d$ ($\mu$M)[a] | FMNAT $IC_{50}(\mu M)$[b] | FMNAT % Res. act. 250 $\mu$M | RFK+FMNAT $IC_{50}(\mu M)$[b] | RFK+FMNAT % Res. act. 250 $\mu$M |
|---|---|---|---|---|---|
| C2 | | | >95 | | 88 ± 2 |
| C3 | 18 ± 8 | 238 ± 7 | 48 ± 3 | 248 ± 3 | 48 ± 5 |
| C5 | 1.7 ± 0.7 | 53 ± 1 | 6.0 ± 1.4 | 83 ± 2 | 14 ± 1.3 |
| C6 | 41 ± 3 | 96 ± 6 | 48 ± 6 | | 57 ± 3 |
| C7 | | | >95 | | 82 ± 7 |
| C9 | 6.4 ± 1.2 | | 56 ± 6 | | 76 ± 3 |
| C18 | 3.0 ± 0.9 | 143 ± 4 | 35 ± 4 | 147 ± 4 | 35 ± 4 |
| C26 | | | 72 ± 6 | | 84 ± 4 |

The table includes dissociation constant ($K_d$) for the compounds altering thermal stability of *Ca*FADS, concentration of compound causing 50% enzyme inhibition ($IC_{50}$) and residual activity at 250 $\mu$M of compound for the FMNAT and RFK+FMNAT activities of *Ca*FADS. Thermal stability and activity experiments were carried out in 20 mM PIPES, pH 7.0, 10 mM MgCl$_2$. *Ca*FADS activities were assayed at 25 ˚C. All samples contained 2% DMSO. (n = 3, mean± SD).

[a] Obtained from differential scanning fluorescence data and

[b] kinetic measurements. For details see the Methods.

 

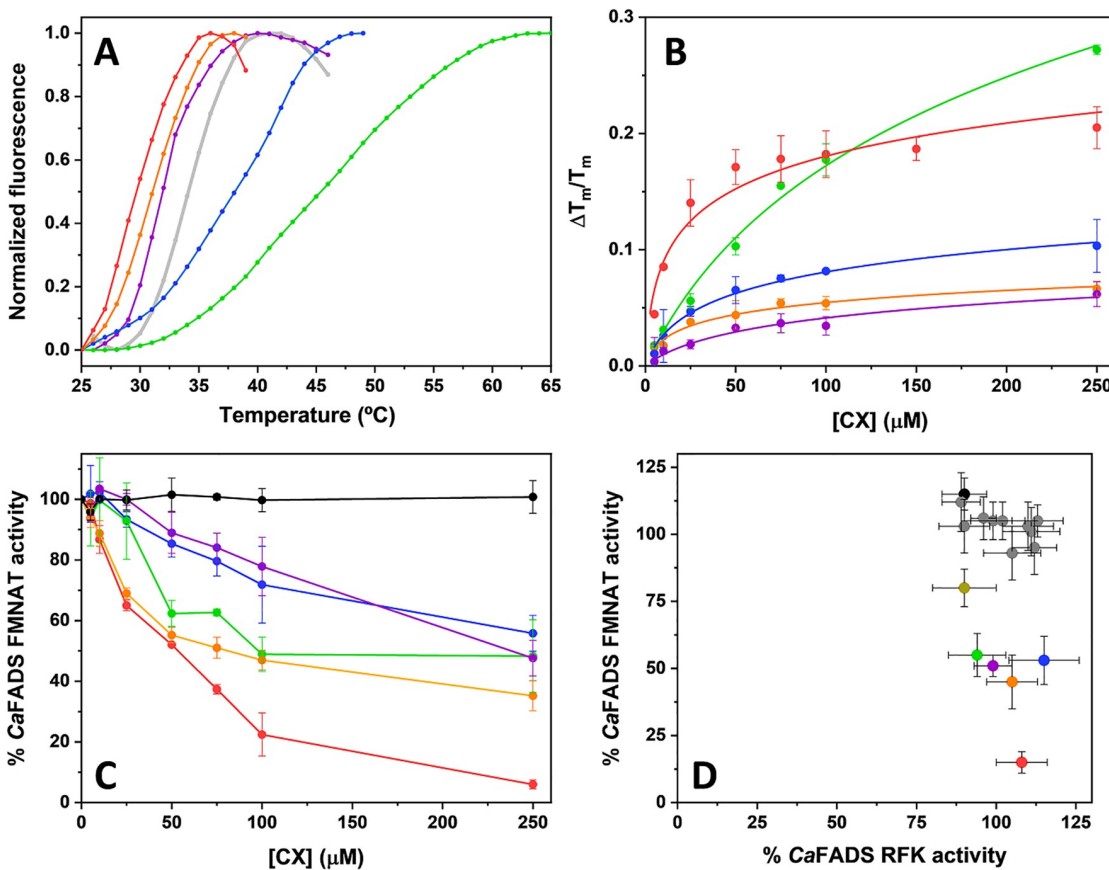

**Fig 6. *In vitro* assessment of VSHs ability to bind and to inhibit *Ca*FADS.** A) Thermal denaturation curve for *Ca*FADS (2 $\mu$M) observed by differential scanning fluorescence and $T_m$ shifts observed in the presence of the compounds at 250 $\mu$M. Thermal stability curves are plotted against the normalized fluorescence signal. Experiments were carried out in 20 mM PIPES, pH 7.0, 10 mM MgCl$_2$, 2% DMSO. B) Dependence of $\Delta T_m$ on the VSH concentration and data fit to [Eq 2](). C) Dose-response curves for the FMNAT activity of *Ca*FADS in the presence of representative VSHs. Experiments performed at 25 °C in 20 mM PIPES, pH 7.0, 10 mM MgCl$_2$, 2% DMSO, with 5 $\mu$M FMN and 50 $\mu$M ATP. Values derived from these representations are included in [Table 2](), such as the IC$_{50}$ and % of remaining activity at 250 $\mu$M of the VSH. D) Comparison of the effects of the VSHs on the RFK and FMNAT activities of *Ca*FADS. All the experiments were carried out at 25 °C, in 20 mM PIPES pH 7.0, MgCl$_2$ (10 mM when assaying FMNAT activity and 0.8 mM when assaying RFK activity) at saturating concentrations substrates and in the presence of 250 $\mu$M of the VSH (2% DMSO, final concentration). Compound color code: Protein in the absence of VSH is shown in light gray, C3 is violet, C5 is red, C6 is green, C9 is blue, C12 is black (shown as control, neither binder nor inhibitor) and C18 is orange. Note that not all molecules are shown in all panels. In panel D, compounds different from the above mentioned are indicated in dark gray and calculated activity percentages are relative to the corresponding ones in absence of compounds. (n = 3, mean ± SD).

well as of the percentage of remaining activity at the highest compound concentration assayed (250 $\mu$M) ([Fig 6C](), third and fourth columns in [Table 2]()). 6 out of the 17 VSHs produced some inhibitory effect on the FMNAT activity. These were compounds C3, C5, C6, C9, C18 and C26. Among them, C5 and C6 yielded IC$_{50}$ values below 100 $\mu$M, C5 IC$_{50}$ = 53±1 $\mu$M and C6 IC$_{50}$ = 96±6 $\mu$M, with C5 inhibiting over 90% the FMNAT activity of *Ca*FADS at the maximal compound concentration assayed. The structure of these compounds is shown in [S5 Fig]().

 *Ca*FADS is a bifunctional enzyme that in addition to the FMNAT N-terminal module holds an RFK C-terminal module that transforms riboflavin (RF) into FMN, producing the substrate of the FMNAT activity. The presence of this second module was not considered in our VS protocol, where only the ATP/FMN binding pocket of the FMNAT module was used as the active site. Nonetheless, since the RFK module also comprises an active site binding adenine and

**Table 3. *In vivo* performance of VS hits.**

| Compound | MIC ($\mu$M)<br>*C. ammoniagenes* | MIC ($\mu$M)<br>*M. tuberculosis* |
|---|---|---|
| C2 | 128 | 256 |
| C3 | >256 | 256 |
| C5 | 32 | 128 |
| C6 | 128 | 128 |
| C27 | 128 | 256 |

VSHs which minimal inhibitory concentration (MIC) against *C. ammoniagenes* and *M. tuberculosis* is lower than 256 $\mu$M.

flavin nucleotides, its RFK activity might also been affected by the VSHs. Therefore, we also evaluated the ability of *Ca*FADS to transform RF into FMN and subsequently into FAD. Fig 6D, compares the effect of the VSHs on both individual activities of *Ca*FADS, RFK and FMNAT, showing that, under the assay conditions, the 17 hits produced minor effects on the RFK activity. In agreement, when evaluating the overall *Ca*FADS activity the effect of the VSHs follows a similar trend to that when individually evaluating the FMNAT activity.

To assess the effect of VSHs on the growth of different bacteria, we determined their MIC (Table 3 and S2 Table). Bacterial cells of *C. ammoniagenes*, *Corynebacterium glutamicum*, *Corynebacterium diphteriae*, *M. tuberculosis*, *M. smegmatis*, *S. pneumoniae*, *E. coli*, *Listeria monocytogenes*, *Pseudomonas aeruginosa*, *Salmonella thyphimurium*, *Staphylococcus aureus* and *Bacillus spp.* were grown in the presence of increasing concentrations of the selected VSHs. Among the VSHs, C2, C5, C6, C18 and C27 produced a detectable inhibition in the growth of *C. ammoniagenes*, being C5 (MIC = 32 $\mu$M) the compound producing the largest antibacterial effect followed by C2 (MIC = 64 $\mu$M). Interestingly, C5 is also the hit exhibiting the lowest $IC_{50}$ for the FMNAT activity of *Ca*FADS (Table 2). The five VSHs exhibiting antibacterial activity against *C. ammoniagenes* also had activity on the other *Corynebacterium* species analyzed, being particularly relevant the effects of C1 and C5 on *C. glutamicum* as well as of C5 and C18 on *C. diphteriae*. Four of these five compounds, C2, C5, C6 and C27, as well as C3, had also antibacterial effect in the growth of *Mycobaterium* species, although they were in general less potent. In addition, C18 and C27 produced moderate MIC values (64 $\mu$M) for *L. monocytogenes* growth, C6 for *S. pneumoniae*, and C27 for *S. aureus*. It is also worth to note the inhibition of *Baccillus spp.* growth caused by C5 and C27 (MIC = 32 $\mu$M).

In general, we observed that the VSHs showing inhibitory activity against *Ca*FADS also inhibit the growing of *Corynebacterium* species. Thus, we can hypothesize that the growing inhibition effect should be caused by the *Ca*FADS inhibition. In addition, those VSHs also inhibit the growing of the *Mycobacterium* species, supporting *Ca*FADS as a representative model of the FADS of *M. tuberculosis*. The fact that the compounds identified through VS have demonstrated some antimicrobial activity is an important result, even when this antimicrobial activity is moderate; historically, potential enzyme inhibitors identified through *in silico* or *in vitro* protein-binding assays are mostly devoted of any antimicrobial activity, due to their inability to cross the high permeability barrier posed by the bacterial envelope [77].

Finally, we evaluated for the effect of VSHs on eukaryotic cell growth and viability. Compounds C2, C5 and C6 were not cytotoxic in HeLa and A549 cell lines, with $IC_{50}$ (concentration of compounds causing the 50% inhibition of the cellular viability) above the maximal concentration evaluated (512 $\mu$M). In contrast, C18 and C27 showed moderate cytotoxicity in both cell lines but only in the 256-512 $\mu$M range, with complete viability being retained at

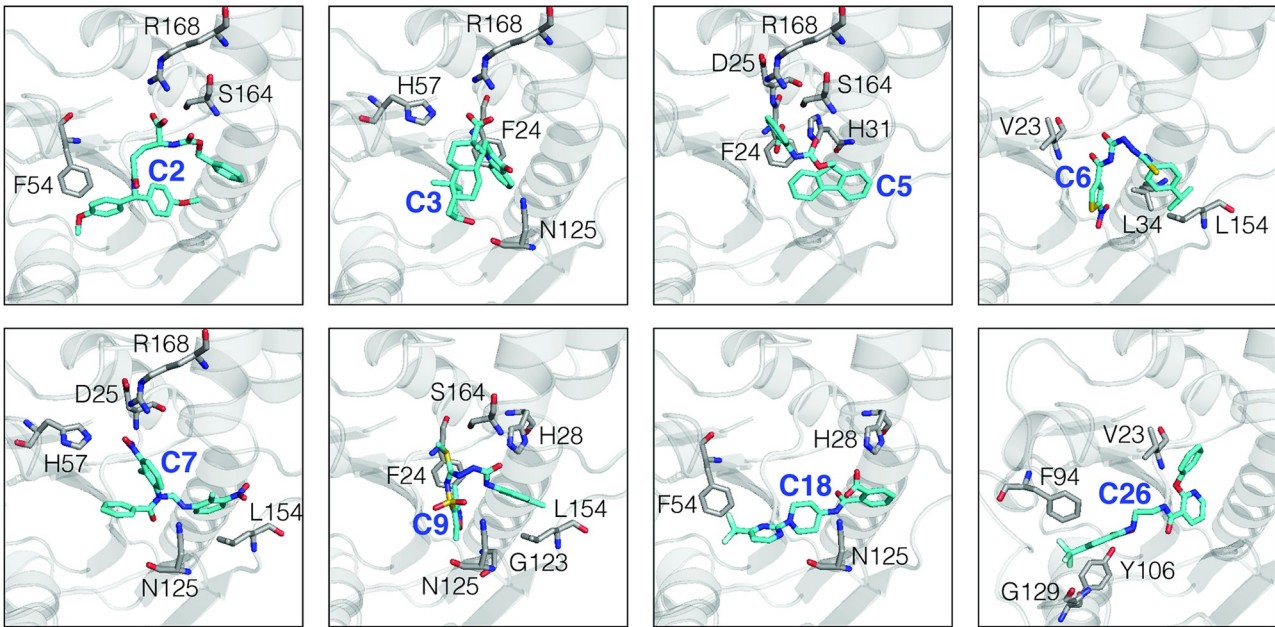

**Fig 7. Docking poses showing the main interactions of the VSHs and *Ca*FADS.** These were obtained with Autodock4.2 and were used as the starting conformation for the MD stage.

lower concentrations. Thus, the compounds do not show high cytotoxic effects against eukaryotic cells, highlighting their potential use against prokaryotic pathogens.

## Binding pocket VSHs interactions

Fig 7 shows the main interactions between the VSHs (from Table 2) and *Ca*FADS. According to the simulations, most hits interact with the ATP binding site (C2, C3, C5, C6, C7 and C9). Only two compounds (C18 and C26) interact with the binding site of both natural substrates (ATP and FMN). Detailed interactions between the VSHs and the *Ca*FADS are represented in S6 Fig. A global view of the VSHs inside the binding pocket is shown in S7 Fig. It is worth to note that all the ligands show direct interactions with key residues in the FMAT binding pocket. Compounds C2, C3, C5, C6, C7, C9 and C18 interact with N125, which is considered the key catalytic residue for the FMNAT activity. In addition, all compounds show direct interactions with highly conserved residues, particularly those responsible for the stabilization of the phosphates of the ATP (H28, H31, H57, S164 and R168).

## Conclusion

We developed a VS protocol that is able to find ligands of an enzyme which does not require previous knowledge of ligands or ligand-receptor structures. The protocol is computationally efficient allowing for the screening of large compound libraries with moderate computational resources. The protocol was implemented over *Ca*FADS, an enzyme that is considered a good model for FADSs of bacterial species that cause tuberculosis and pneumonia [40, 41].

The VS protocol involves a funnel-like strategy with filtering stages that increase in accuracy. In the first stage, we used the flexi-pharma method [31], a pharmacophore filtering strategy with ligand-free receptor conformations from MD. In the second stage, we used a consensus docking strategy to combine the results from different docking programs using the exponential consensus ranking (ECR) method [43]. In the third stage of MD-ranking, we

developed a new score for the ligand's flexibility using a Morse potential. This score is combined with other scoring functions using the ECR method over the MD ensemble.

The protocol was optimized and validated over an experimentally-tested compound library with known ligands of the *Ca*FADS. We implemented the VS strategy over a unexplored compound library, resulting in a list of 17 compounds that were tested experimentally. Notably, we discovered five compounds able to bind to the *Ca*FADS. One of these compounds shows significant inhibition of the FMNAT activity of *Ca*FADS. In comparison to previous work [41], the computational protocol gives an enrichment of around 8 for the experimental stage. In addition, some of the new compounds show growth inhibitory activity against *Corynebacterium*, *Mycobacterium* or *Streptococcus* species, supporting the use of the integrative VS protocol for the initial stages of drug discovery.

Although our final results show a good experimental enrichment, we note that the generalization of the entire protocol (shown in Fig 1) is still to be optimized, and validated for multiple systems with diverse active sites.

## Supporting information

**S1 Fig. Example of the Flexi-pharma pharmacophore mapping.** The green, red and gray spheres represent hydrogen-bond acceptor, negatively charged and aromatic features, respectively.
(TIF)

**S2 Fig. Enrichment plot for the different ECR-docking strategies.** The violet line shows the EP from ECR combination of Autodock4.2, Vina and Smina docking results using the best pose from each program (as was done in ref. [43]). We also studied the outcome when using the best pose for each molecule from the different programs: from Autodock4.2 (black), Vina (green) and Smina (blue), then re-scored it with Autodock4.2, Vina, Vinardo and CYscore, and these new scores were combined using an ECR methodology. We find that using the Autodock4.2 pose and re-scoring it with the other programs produces the best outcome.
(TIF)

**S3 Fig. Funnel-like protocol implemented for the Maybridge compound library.** The number of filtered molecules is shown on the left. The computational protocol has several stages: first, a pharmacophore-based VS (flexi-pharma), then ECR-Docking, afterwards two MD stages (that depended on the simulation time) were used for ranking the compounds with a Morse-based score and an ECR combination of scoring functions. In the physico-chemical stage, we assessed a range of properties for the 30 best VS-ranked compounds that relate to their potential drug-likeness (S1 Table), as well as their commercial availability, selecting 17 compounds for the experimental assays. 5 compounds were found to be ligands of *Ca*FADS.
(TIF)

**S4 Fig. Ranks for the individual scoring functions and the ECR method for the MD-ranking stage 2 using 300 molecules filtered from the Maybridge compound library.** The best ranked molecules by the ECR are also well ranked for the majority of the programs but not necessarily for all. The top 30 molecules given by the ECR are selected for the following stage.
(TIF)

**S5 Fig. Structure of the active VSHs towards *Ca*FADS or the organisms tested.** C3, C5, C6, C9 and C18 are able to bind to *Ca*FADS. C3, C5, C6 and C18 cause 50% of FMNAT activity inhibition ($IC_{50}$) at concentration lower than 250 μM. C3, C5 and C18 cause 50% RFK-

FMNAT activity inhibition ($IC_{50}$ at concentration lower than 250 μM. C3, C5, C6, C9, C18 and C26 have FMNAT residual activity < 95% at 250 μM of compound. C2, C3, C5, C6, C7, C9, C18 and C26 have RFK+FMNAT residual activity < 95% at 250 μM of compound. C2, C5, C6 and C27 have MIC values lower than 256 μM against *C. ammoniagenes*, and or C2, C3, C5, C6 and C27 *Mycobacterium*species.
(TIF)

**S6 Fig. 2D representation of interactions between VSHs and *Ca*FADS found in the docking stage.** The Autodock pose obtained in the docking stage was used to observe the interactions with *Ca*FADS. Compounds C2, C3, C5, C6, C7 and C9 interact with the ATP binding site and compounds C18 and C26 show interactions with the binding site of both ATP and FMN. Almost all compounds interact with key residues in the binding pocket: H28, H31, H57, N125, S164 and R168.
(TIF)

**S7 Fig. Global view of the *Ca*FADS receptor with VSHs.** A global view of the receptor structure with the superposition of the Autodock poses of the VSHs compounds. The docked compounds cover a wide range of the receptor binding pocket.
(TIF)

**S1 Text. Description of the different ECR-docking FADS optimization strategies.**
(PDF)

**S1 Table. Summary of properties for the *Ca*FADS best ranked VS compounds.** The table shows the VS rank, the Zinc and Maybridge Codes, the names and summary of physico-chemical criteria (values at pH 7.0) to evaluate their potential drug-likeness. Preferred criteria values are indicated on the top in green. Favorable criteria for each compound are highlighted in a green background, those in the limit are in a yellow background and those violating the criteria are in a red background. Compounds selected for experimental evaluation as virtual screening hits (VSH) are highlighted in red font in the first four columns.
(PDF)

**S2 Table. Minimal Inhibitory Concentration (MIC) of VSHs against different microorganisms.** Compounds were, in most cases, assayed in the 0-256 μM concentration range. In some cases, they were assayed only in the 0-64 μM concentration range, and if no effect was observed in these cases > 64 is shown. Best performing compounds are colored from red, orange, yellow to green.
(PDF)

**S3 Table. Bacterial strains tested for VSH antibacterial activity.**
(PDF)

## Acknowledgments

The authors would like to acknowledge the use of Servicios Generales de Apoyo a la Investigación-SAI, Universidad de Zaragoza. We thank M. Minjarez and M. Martinez-Júlvez for their collaboration in evaluating physico-chemical properties of VS hits. The authors would also like to acknowledge Dr. Claudio Cavasotto for insightful discussions about the Flexi-pharma method. Some computations were performed in a local server with an NVIDIA Titan X GPU. P.C. gratefully acknowledges the support of NVIDIA Corporation for the donation of this GPU.

## Author Contributions

**Conceptualization:** Isaias Lans, Ernesto Anoz-Carbonell, José Antonio Aínsa, Milagros Medina, Pilar Cossio.

**Data curation:** Isaias Lans, Ernesto Anoz-Carbonell, Karen Palacio-Rodríguez, José Antonio Aínsa, Milagros Medina, Pilar Cossio.

**Formal analysis:** Isaias Lans, Ernesto Anoz-Carbonell.

**Funding acquisition:** José Antonio Aínsa, Milagros Medina, Pilar Cossio.

**Investigation:** Isaias Lans, Ernesto Anoz-Carbonell, Karen Palacio-Rodríguez, José Antonio Aínsa, Milagros Medina, Pilar Cossio.

**Methodology:** Isaias Lans, Ernesto Anoz-Carbonell, Karen Palacio-Rodríguez.

**Resources:** José Antonio Aínsa, Milagros Medina, Pilar Cossio.

**Software:** Isaias Lans.

**Supervision:** José Antonio Aínsa, Milagros Medina, Pilar Cossio.

**Validation:** Isaias Lans, Ernesto Anoz-Carbonell, Karen Palacio-Rodríguez, José Antonio Aínsa, Milagros Medina, Pilar Cossio.

**Visualization:** Isaias Lans, Ernesto Anoz-Carbonell, Karen Palacio-Rodríguez.

**Writing – original draft:** Isaias Lans, Ernesto Anoz-Carbonell, Karen Palacio-Rodríguez, José Antonio Aínsa, Milagros Medina, Pilar Cossio.

**Writing – review & editing:** Isaias Lans, Ernesto Anoz-Carbonell, Karen Palacio-Rodríguez, José Antonio Aínsa, Milagros Medina, Pilar Cossio.

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
