## [Decision Letter · Decision Letter 0]

13 Jun 2020

Dear Dr Cossio,

Thank you very much for submitting your manuscript "In silico discovery and biological validation of ligands of FAD synthase, a promising new antimicrobial target" for consideration at PLOS Computational Biology. As with all papers reviewed by the journal, your manuscript was reviewed by members of the editorial board and by several independent reviewers. The reviewers appreciated the attention to an important topic. Based on the reviews, we are likely to accept this manuscript for publication, providing that you modify the manuscript according to the review recommendations.

Sincerely,

Alexander MacKerell

Associate Editor

PLOS Computational Biology

Rob De Boer

Deputy Editor

PLOS Computational Biology

[LINK]

Reviewer's Responses to Questions

**Comments to the Authors:**

Reviewer #1: The authors propose a virtual screening cascade for unexplored targets and/or targets without co-crystallized ligands. They apply this strategy to the FAD synthase, calibrate the protocol with some known compounds and then run the workflow that includes different steps with MD simulation, rescoring, consensus scoring with a method that the group has developed … They then select some molecules coming from this protocol and test the molecules experimentally and identify several hits. This is in my opinion a very interesting study. Yet, I have several points and questions below that should be very easy to clarify but I think missing in the present version of the manuscript.

1) Open Babel uses simple rules for the protonation of essentially amino acids, which can be perfectly fine for some chemical groups on some compounds, but for many small drug-like molecules, the package fails. The authors should explain what is the protonation state of the compounds as it has obviously an impact for scoring and for MD simulation. We have on our side used extensively Open Babel and we know that the protonation is wrong for many many groups… Thus the question. Also, how do the authors deal with standardisation of the compounds ? One Nitro group can be neutral, or charged… it depends the way it is written in the file…

The point about protonation state of the compounds appears in the MD method section, maybe it belongs to the Docking section?

2) The Flexi-Pharma method is not published but used in the present study but not all the authors of the Flexi-Pharma method are involved in this study, why (eg., Prof Cavasotto is not co-author here) ?

The authors wrote in the section Flexi-pharma: First, an MD simulation of ligand-free receptor is performed. But there is no explanation of how this is performed? how long etc… with constraints etc…I do not think this is in the supplement?

With flexibility in the definition of the pharmacophore, don’t the authors see an increase of noise compared to a much smaller set of pharmacophores? In docking small compounds, using pre-generated conformational ensemble, one tends to observe improvements in hit recovery only if there are not too starting conformation of the receptor. As soon as the number of receptor reaches 5 or so, there is no improvements but noise, meaning almost all compounds fit in the pocket. Is it the situation here for the definition of the pharmacophore models ?

3) With regard to molecular docking, the authors should explain the type of compound input files that they used for Smina (a single mol2 or sdf…file ? or pdbqt?) as this can change significantly the scores for some molecules.

4) The authors should show some docked poses, show and explain the properties of the binding pocket (large or small cavity, ions, cofactor or not…).

5) Several compounds have a nitro group (or two, this is usually not very welcome in a drug discovery project but tolerated at the hit stage) or somewhat a negatively charge group in the ligand. What is the rational with regard to the binding pocket? If this charge or interaction is that important, could it be the key reason of the success of the MD and/or pharmacophore and/or scoring? In a more hydrophobic pocket without this interaction, i assume maybe there is a key salt-bridge, the situation could be very different and what is happening then with the screening protocol ? The authors should explain how much this approach is likely to generalise to other types of pocket.

Reviewer #2: The work described in this article is scientifically sound and looks carefully done in all its parts. Due to my specific expertise, my comments are limited and focused on the experimental part of the study for which I have no particular criticism. The article is written in a concise and and very clear style. I just recommend the Authors to carefully check and fix some minor formal typos such as the lack of a space between numbers and '°C' unit.

Reviewer #3: The article describing the computational approach that led the authors identify FADS inhibitor. Here authors developed new screening protocol and successfully employed to identify hit compounds in µM range that has potential to become anti-microbial therapy. The manuscript carefully written and presented. The protocol differ from classical virtual screening, where the last filter is usually docking. Here, authors exercised molecular dynamics to pick 17 compounds. Selecting the compounds include state-of-the-art physicochemical properties into consideration that increase drug-likeness.

I have some of the following concerns (minor) to be addressed before acceptance.

1) Flexi-pharma mapping on the screened compounds are not shown.

2) Although there is a reference to ECR method, I don’t see some details in the methods as well as the ECR versus the individual scoring function. It helps to see them side-by-side in a table.

3) Hit molecule interaction in the receptor context is not available, I will recommend to present that figure.

**Have all data underlying the figures and results presented in the manuscript been provided?**

Reviewer #1: Yes

Reviewer #2: Yes

Reviewer #3: Yes

PLOS authors have the option to publish the peer review history of their article (what does this mean?). If published, this will include your full peer review and any attached files.

Reviewer #1: No

Reviewer #2: No

Reviewer #3: No
---

## [Editor Report · Decision Letter 1]

9 Jul 2020

Dear Dr Cossio,

We are pleased to inform you that your manuscript 'In silico discovery and biological validation of ligands of FAD synthase, a promising new antimicrobial target' has been provisionally accepted for publication in PLOS Computational Biology.

Best regards,

Alexander MacKerell

Associate Editor

PLOS Computational Biology

Rob De Boer

Deputy Editor

PLOS Computational Biology

---

## [Editor Report · Acceptance letter]

7 Aug 2020

PCOMPBIOL-D-20-00622R1 

In silico discovery and biological validation of ligands of FAD synthase, a promising new antimicrobial target

Dear Dr Cossio,

I am pleased to inform you that your manuscript has been formally accepted for publication in PLOS Computational Biology. Your manuscript is now with our production department and you will be notified of the publication date in due course.

With kind regards,

Matt Lyles
